# Hydrogel-Based Strategies for the Prevention and Treatment of Radiation-Induced Skin Injury: Progress and Mechanistic Insights

**DOI:** 10.3390/biomimetics10110758

**Published:** 2025-11-11

**Authors:** Yinhui Wang, Huan Liu, Yushan He, Mei Li, Jie Gao, Zongtai Han, Jiayu Zhou, Jianguo Li

**Affiliations:** CNNC Key Laboratory on Radio-Toxicology and Radiopharmaceutical Preclinical Evaluation, CAEA Center of Excellence on Nuclear Technology Applications for Nonclinical Evaluation for Radiopharmaceutical, Shanxi Key Laboratory of Drug Toxicology and Preclinical Studies for Radiopharmaceutical, Division of Radiation Medicine and Environmental Medicine, China Institute for Radiation Protection, Taiyuan 030006, China; wangyinhuii@foxmail.com (Y.W.); happyliusxu@foxmail.com (H.L.); cen_98@foxmail.com (Y.H.); limei@cirp.org.cn (M.L.); gaojie@cirp.org.cn (J.G.); hanzongtai@foxmail.com (Z.H.); zjy00720@sina.com (J.Z.)

**Keywords:** hydrogel, radiation-induced skin injury, natural polymers, synthetic polymers, hybrid hydrogels, smart hydrogels, mechanisms

## Abstract

Radiation-induced skin injury (RISI) is one of the most common complications of radiotherapy, severely compromising patients’ quality of life. However, no standardized treatment has yet been established. Owing to their high water content, three-dimensional porous structure, excellent biocompatibility, and tunable functionalization, hydrogels have emerged as promising candidates for both the prevention and treatment of RISI. This review provides a comprehensive overview of recent advances in hydrogel-based interventions for RISI, with particular focus on material classifications and underlying mechanisms. Mechanistically, hydrogels facilitate tissue repair through multiple synergistic pathways, including antioxidation, anti-inflammation, angiogenesis, and tissue remodeling. Understanding these mechanisms not only provides a theoretical basis for the rational design of next-generation wound dressings but also enhances the translational potential of hydrogels in clinical radiotherapy. With the convergence of materials science, radiation medicine, and pharmaceutical innovation, hydrogels are poised to redefine therapeutic strategies for RISI and accelerate their clinical implementation.

## 1. Introduction

With the continuous advancement of radiation medicine, radiotherapy (RT) has become a primary and indispensable approach in cancer treatment. Although radiation damages both malignant and normal cells, normal cells generally possess a stronger capacity for self-repair, thereby maintaining their functional integrity [1]. The improvement in patient survival has drawn greater attention to treatment-related sequelae, which often impose long-term physical and psychosocial burdens on patients. During RT, a fraction of low-energy radiation is absorbed by the skin, aggravating local reactions and resulting in radiation-induced skin injury (RISI). RISI is among the most common complications of radiotherapy, affecting 85–95% of patients receiving standard RT, with particularly high incidence in those treated for head and neck, breast, or skin cancers [2]. The severity of RISI varies with the irradiated region and may present as erythema, edema, pigmentation changes, or desquamation of varying intensity [3]. Clinically, RISI exhibits a graded progression: initial erythema and dry desquamation may advance to moist desquamation, and in severe cases, skin ulceration [4]. However, no standardized therapeutic regimen for RISI has yet been established.

### 1.1. Mechanisms and Clinical Background of RISI

RISI can be classified as acute or chronic. Acute RISI typically develops within 90 days after ionizing radiation. According to the Common Terminology Criteria for Adverse Events (CTCAE v3.0), its clinical severity ranges from Grade 1 to Grade 4. Grade 1 is characterized by dry desquamation and generalized erythema; Grade 2 presents with intense erythema or patchy moist desquamation, and when the cumulative dose reaches 40 Gy or higher, moist desquamation frequently appears in skin folds. Grade 3 is defined by extensive moist desquamation beyond skin folds, while Grade 4 involves ulceration, bleeding, and necrosis. Pathological manifestations include edema, endothelial cell changes and other epidermal and dermal cell changes, such as inflammatory infiltration, apoptosis and necrosis caused by lymphocytes and cytokines [5].

Chronic RISI, by contrast, typically emerges months to years after radiation exposure, presenting as chronic ulcers, fibrosis, telangiectasia, secondary skin cancers, or radiation-induced keratosis. Histopathological features include alterations in cell density, fibrotic tissue deposition, pigmentation, and vascular distribution [6].

The pathogenesis of RISI involves a complex cascade of biological responses. Ionizing radiation directly induces various forms of cell death, such as mitotic catastrophe, while also triggering local inflammatory reactions through the release of bioactive mediators from both immune and non-immune cells [7]. Multiple signaling pathways are activated in the skin, particularly those related to cytokine secretion, redox imbalance, and impaired angiogenesis [8,9,10]. Persistent accumulation of reactive oxygen and nitrogen species further exacerbates tissue injury, suppresses DNA repair, and disrupts both barrier integrity and tissue remodeling. Moreover, signaling molecules such as TGF-β drive abnormal fibroblast activation in the chronic phase, serving as a major force in fibrosis development [11]. Thus, the progression of RISI is a dynamic process evolving from acute inflammation to chronic oxidative damage and fibrosis [12], ultimately compromising the structural stability and regenerative capacity of the skin. Figure 1 illustrates the alterations in skin cells and cellular signaling following radiotherapy [13].

It is worth noting that radiation-induced damage not only manifests as acute inflammation and chronic fibrosis but also severely disrupts the natural wound-healing process of the skin. The underlying mechanisms involve microvascular destruction, extracellular matrix abnormalities, and depletion of reparative cells, resulting in local hypoxia and diminished regenerative potential. Meanwhile, the persistent activation of inflammatory and fibrotic mediators (such as TNF-α, IL-6, TGF-β, and CTGF) and adhesion molecules (VCAM-1, ICAM-1) drives progressive tissue injury and abnormal scar formation. Consequently, radiation exposure affects nearly all phases of wound repair—including hemostasis, inflammation, proliferation, and remodeling—ultimately leading to fibrotic scarring.

Moreover, recent studies have demonstrated that radiation-induced senescence of keratinocytes plays a critical role in the chronic progression of RISI [14]. Following radiation exposure, epidermal keratinocytes acquire a senescence-associated secretory phenotype (SASP) after entering stable growth arrest, continuously releasing cytokines, chemokines, and proteases that activate immune cells and disrupt skin-barrier homeostasis. This process is accompanied by upregulation of cell-cycle inhibitory proteins (p16^INK4a and p21^WAF1) and loss of lamin B1, leading to amplified inflammatory signaling and impaired tissue regeneration (Figure 2).

Clinically, the key to treating RISI lies in the concurrent regulation of the inflammatory response, oxidative stress, and tissue remodeling. Traditional therapies primarily focus on symptomatic relief and lack targeted modulation of the underlying pathological mechanisms. In cases of chronic RISI, pharmacological agents such as pentoxifylline, recombinant human epidermal growth factor (rhEGF), and adipose-derived stem cells (ASCs) [15,16,17] have demonstrated certain therapeutic benefits in clinical practice. In contrast, the management of acute RISI mainly depends on topical corticosteroids (e.g., mometasone furoate and betamethasone) [18,19] and emollients (e.g., triethanolamine cream) [20], which alleviate inflammation and discomfort by suppressing the release of proinflammatory cytokines. However, these regimens are largely symptomatic treatments that may impair the skin barrier and fail to prevent chronic fibrotic progression. The commonly used drugs and mechanism-oriented therapeutic strategies for RISI are summarized in Table 1.

With the deepening understanding of the molecular mechanisms underlying RISI, therapeutic research has gradually shifted from nonspecific symptomatic management to the development of mechanism-based radioprotective agents. One of the earliest representative drugs is Amifostine, which scavenges radiation-induced reactive oxygen species (ROS) to prevent DNA damage and suppresses treatment-related inflammatory pathway activation, thereby reducing the incidence of radiation-induced skin injury [41,42,43]. This mechanism-driven protective strategy has laid the groundwork for the development of subsequent dual-target antioxidant and anti-inflammatory agents. In recent years, bioactive approaches such as mesenchymal stem cell (MSC) therapy have further advanced this concept, achieving more comprehensive tissue protection and repair by synergistically attenuating oxidative stress and inflammatory responses [38,44]. Building on this foundation, emerging material-based strategies—particularly hydrogel dressings—integrate physical shielding, controlled drug delivery, and wound-healing promotion, offering a new therapeutic paradigm for the comprehensive protection and regeneration of RISI.

### 1.2. Hydrogel Carrier Materials and Their Multifaceted Potential

Current radioprotective agents for the skin primarily rely on chemical repair mechanisms, and no physical protective material has yet been developed that can effectively shield low-energy radiation. As a polymeric network system [45], hydrogels possess exceptional water-retention capacity, allowing them to serve as benign physical barriers that mitigate the effects of low-energy ionizing radiation, while remaining adherent to tissue surfaces for extended periods. Moreover, they can be engineered to encapsulate growth factors, stem cells, and other bioactive substances [46,47,48], ensuring sustained release, enhanced bioavailability, and combined protective and therapeutic effects. These attributes position hydrogels as strong candidates for the next generation of radioprotective agents with considerable developmental promise. As shown in Figure 3, hydrogels exert synergistic effects in protecting and repairing RISI through multidimensional mechanisms—including physical shielding, chemical repair, and biological healing—offering a novel concept that integrates protection and therapy.

Hydrogels are typically composed of hydrophilic polymers that are chemically or physically cross-linked to form a three-dimensional network structure [50], characterized by their ability to swell upon exposure to water. This unique architecture not only imparts favorable mechanical properties but also closely mimics the extracellular matrix, thereby conferring excellent tissue compatibility. These characteristics render hydrogels highly attractive as drug carriers, tissue-engineering scaffolds, and beyond [51]. In addition, their high water content makes them outstanding wound dressings, capable of providing an ideal moist environment for tissue repair [52]. Importantly, the multifunctional drug delivery capacity of hydrogels allows for the incorporation of diverse bioactive components tailored to clinical needs. They can be administered by injection for precise delivery or applied topically to wounds by incorporating anti-radiation agents, offering flexible and versatile solutions for complex injuries such as RISI.

This review summarizes recent advances in the use of hydrogels for the prevention and treatment of RISI, focusing on their classification by material type and mechanisms of action. Current evidence indicates that hydrogels not only provide a moist healing environment and physical barrier but also exert additional therapeutic effects—including antioxidation, anti-inflammation, angiogenesis, and tissue repair—through the incorporation of active molecules or cells. Thus, hydrogels offer dual protective and therapeutic benefits. However, high-performance hydrogel products specifically developed for RISI remain in the research and development phase, and clinical translation is still constrained by challenges related to stability, safety, large-scale manufacturing, and regulatory approval. Although hybrid and intelligent hydrogels have demonstrated promising potential for precise drug delivery and stimuli-responsive regulation [53], their complex fabrication processes and long-term safety require further validation. Looking ahead, differentiated intervention strategies should be designed according to the stages of RISI progression. Through interdisciplinary collaboration among materials science, radiation medicine, and bioengineering, hydrogels hold promise for translation from laboratory research to clinical application, providing more reliable solutions for the prevention and treatment of RISI.

## 2. Research Progress on the Mechanisms of Hydrogels in RISI

The unique materials science and biological properties of hydrogels endow them with broad potential for diverse applications. Natural polymers (such as chitosan, hyaluronic acid, and gelatin) and selected synthetic polymers (such as PEG and PVA) exhibit excellent biocompatibility and biodegradability, ensuring high safety and suitability for long-term use [54]. The combination of porous architecture and modifiable chemical groups further enables hydrogels to efficiently encapsulate drugs, antioxidants, gene fragments, protein factors, or cells, thereby achieving sustained release and localized delivery while minimizing systemic side effects [55]. In addition, their adjustable mechanical properties, capacity for functionalization and intelligent responsiveness [56], as well as barrier and antibacterial functions [57], collectively expand their biomedical utility. These fundamental characteristics have allowed researchers to design a wide array of hydrogel-based therapeutic strategies tailored to different pathological stages of RISI.

For clarity of discussion, this review classifies hydrogels into four major categories: (i) natural-origin hydrogels and their composites, primarily derived from polysaccharide or protein matrices; (ii) synthetic polymer hydrogels and their composites, highlighting the tunability and functional versatility of engineered polymers; (iii) hybrid hydrogel systems that integrate multiple structural or functional domains; and (iv) smart hydrogels designed to capture cross-disciplinary innovations and respond to external stimuli.

### 2.1. Natural-Origin and Their Composite Hydrogels

Currently, natural hydrogels applied in RISI are primarily derived from polysaccharide-based materials, protein-based materials, and decellularized tissue matrices. Their preparation generally relies on physical or chemical crosslinking, sometimes combined with composite modification. Owing to their excellent biocompatibility, biodegradability, and low cytotoxicity [58], these hydrogels are widely used in tissue repair research, including RISI.

#### 2.1.1. Polysaccharide-Based Materials

Polysaccharides are among the most important natural polymers for hydrogel construction. They are widely utilized in biomedical materials owing to their renewability, biocompatibility, and biodegradability. The abundant functional groups—such as hydroxyl, carboxyl, and amino moieties—present in polysaccharide molecules provide multiple chemical modification, cross-linking, and active-molecule loading sites, enabling the formation of three-dimensional network structures with excellent water absorption, softness, and tunable physicochemical properties. As illustrated in Figure 4, polysaccharide-based materials can be derived from plants or microorganisms, exhibiting multiple advantages including controlled drug release, targeted delivery, protection of sensitive bioactive components, and improved bioavailability [59]. These hydrogels have shown remarkable potential in maintaining a moist wound environment, attenuating inflammatory responses, and promoting tissue regeneration.

Through structural design and functional modification, polysaccharide-based hydrogels can act not only as efficient carriers for drugs or cells, but also as multifunctional therapeutic platforms capable of multidimensional repair of RISI by modulating oxidative stress, inflammatory responses, and angiogenesis. Based on recent research advances, Table 2 summarizes the representative materials, key molecular characteristics, and principal biological roles of polysaccharide-based hydrogels in RISI repair, with emphasis on their antioxidant, anti-inflammatory, and tissue-remodeling functions.

Hyaluronic acid (HA) is a natural glycosaminoglycan composed of glucuronic acid and N-acetylglucosamine, abundantly present in mammalian connective tissues. HA hydrogels contain numerous carboxyl and hydroxyl groups that interact with water molecules to form a stable hydration network. This structure provides strong water-retention capacity and attenuates low-energy X-ray deposition in the skin and surrounding tissues. In addition, HA exhibits moisturizing, anti-inflammatory, and collagen-promoting properties. When combined with bioactive molecules such as ergothioneine, deferoxamine (DFO), or retinoic acid (RA), its therapeutic effects are enhanced. Ergothioneine-loaded HA hydrogels effectively scavenge radiation-induced ROS, thereby alleviating oxidative stress, inhibiting apoptosis, and reducing inflammatory infiltration [73]. DFO stimulates angiogenesis by mimicking hypoxia and upregulating VEGF, while RA promotes fibroblast and hair follicle stem cell proliferation, accelerates wound coverage, and supports scar-free follicle regeneration [74]. Collectively, HA hydrogels act synergistically by reducing oxidative stress, inhibiting apoptosis, suppressing inflammation, promoting angiogenesis and tissue repair, and ultimately improving the healing quality and functional recovery of irradiated skin.

Alginate, a polysaccharide extracted from brown algae, consists of β-1,4-D-mannuronic acid (M) and α-1,4-L-guluronic acid (G) residues linked in varying proportions [75]. The carboxyl groups on alginate chains enable ionic crosslinking with multivalent cations (e.g., Ca^2+^, Ba^2+^, Sr^2+^), generating three-dimensional hydrogel networks. Traditional alginate dressings form a moist wound barrier, absorb exudates, and optimize the microenvironment via ion exchange while preventing secondary infection, thereby facilitating tissue repair [76]. Building on these properties, researchers have developed functionalized alginate hydrogels. For instance, double-network hydrogels (SPM) doped with MoS_2_ nanosheets alleviate oxidative stress by removing excess ROS, preserve epidermal integrity, and reduce inflammatory infiltration, while restoring normal collagen architecture [77]. Similarly, IFI6-PDA@GO/SA hydrogels suppress ROS and NLRP3 expression, reduce keratinocyte apoptosis, and activate HSF1-mediated heat shock responses. These effects enhance stress protection, promote immune activation (CD4^+^/CD8^+^ T cells, NK cells, M1 macrophages), and stimulate angiogenesis and migration, collectively accelerating wound repair [78]. Poly (AAm-SA-DTPA) (PASD) hydrogels [79] further improve wound healing by enhancing TGF-β1 secretion, modulating fibrosis and follicle regeneration, and reducing MDA levels to limit oxidative injury.

Chitosan, a natural alkaline polysaccharide composed of N-acetyl-D-glucosamine units linked by β-1,4-glycosidic bonds, contains amino groups that impart positive charge and enable the formation of hydrogels via electrostatic interactions with negatively charged polymers. Chitosan hydrogels are widely applied as medical dressings. They form a moist protective layer, provide antibacterial barriers, and gradually release collagen and growth factors to stimulate cell proliferation, improve local perfusion, and promote wound healing [80]. Functional modifications have further expanded their therapeutic potential. For example, CSGA/ODex hydrogels scavenge free radicals via phenolic hydroxyl groups, alleviating oxidative stress, while chitosan’s cationic groups disrupt bacterial membranes, conferring broad-spectrum antibacterial activity [81]. CMC/CS/CBD hydrogels incorporate cannabidiol, enhancing selective immune regulation: downregulating pro-inflammatory cytokines, inducing M2 macrophage polarization, and promoting angiogenesis, collagen deposition, and ECM remodeling. Concurrently, they inhibit MMP activity and upregulate adhesion molecules through the MAPK pathway, thereby improving tissue stability and regeneration [82].

Multi-polysaccharide composite hydrogels have also been explored. For example, a balanced-charge hydrogel (AHP-Cur/EGCG) formed by alginate, HA, and polylysine exhibits anti-fouling and antioxidant properties, further enhanced by sustained release of curcumin and EGCG [83]. Mechanistically, this hydrogel upregulates IL-10 and CD31 to promote angiogenesis, while suppressing TNF-α, IL-1β, and IL-6 to mitigate inflammation in irradiated skin. Wang et al. [53] further designed methacrylic anhydride (MA)-modified chitosan/gelatin hydrogels loaded with epigallocatechin gallate (EGCG). The polyphenolic EGCG reduced ROS, enhanced endothelial cell function, and promoted angiogenesis via VEGF, Ang-II, and PDGF-BB upregulation. It also activated autophagy pathways (Beclin-1, LC3) and suppressed inflammatory mediators (NF-κB, MCP-1, PTX-3), thereby alleviating oxidative stress, restoring endothelial function, promoting neovascularization, and accelerating the structural and functional recovery of irradiated skin.

#### 2.1.2. Protein-Based Materials

At present, protein-derived hydrogels have been extensively studied and are recognized for their strong potential in hemostasis and wound repair. As shown in Figure 5, various protein-based materials with hydrogel-forming capability are not only applied in wound repair but also exhibit broad applicability in the regeneration of diverse tissues, including bone, cartilage, myocardium, and nerve tissues [84]. Gelatin, silk fibroin, collagen, elastin, and fibrin are among the most commonly used structural proteins for hydrogel fabrication.

In RISI repair, the key advantage of protein-based hydrogels lies in their ability to rapidly form a soft yet stable support network at the injury site, characterized by excellent fluid absorption and biodegradability. Moreover, owing to the abundant functional groups (e.g., amino, carboxyl, and hydroxyl) present in protein molecules, these hydrogels can interact with platelet membranes or plasma proteins via electrostatic and hydrogen bonding, thereby promoting platelet adhesion, aggregation, and prothrombin activation, ultimately achieving rapid hemostasis [85]. These properties confer unique advantages to protein-based hydrogels in both bleeding control and subsequent tissue regeneration.

Different protein-based materials, owing to their distinct molecular structures and physicochemical characteristics, play differential biological roles in RISI repair (Table 3). Among them, gelatin, silk fibroin, and collagen are the most extensively studied representative materials.

According to Xin Huang et al. [99], gelatin-based hydrogels can rapidly arrest bleeding and maintain a moist microenvironment during the early repair phase, thereby reducing exudate loss and preventing secondary infection. Moreover, small peptide fragments released during gelatin degradation exert anti-inflammatory activity, suppressing pro-inflammatory cytokine expression, mitigating local inflammation, and accelerating the transition from the inflammatory phase to the proliferative phase. During proliferation, gelatin hydrogels provide an excellent scaffold for fibroblast, keratinocyte, and endothelial cell adhesion and migration, thereby enhancing cell proliferation, capillary angiogenesis, and granulation tissue formation. These processes are accompanied by upregulation of angiogenic factors such as VEGF, which further improves perfusion and nutrient delivery. In the remodeling phase, gelatin-based hydrogels promote the conversion of type III collagen into type I collagen, increase collagen fiber density and maturity, and improve the mechanical stability of new tissue. They also accelerate epithelialization and basal layer reconstruction, rendering regenerated skin structurally and functionally closer to normal, thereby shortening healing time.

To further enhance therapeutic performance, researchers have compounded protein-based hydrogels with other natural or synthetic components. For example, silk fibroin exhibits unique mechanical and biological properties, including high tensile strength, low thrombogenicity, and minimal inflammatory response [100]. When combined with perfluorocarbon emulsions, silk fibroin hydrogels improve local hypoxia [101]. Collagen-based hydrogels derived from fish scale extracts [102], enriched in amino acids, enhance antioxidant capacity, reduce ROS accumulation, and mitigate oxidative stress. Their intrinsic structural strength also provides mechanical support against deformation. Additionally, drug- or anti-inflammatory molecule-loaded protein hydrogels achieve localized sustained release, suppress inflammation, and accelerate tissue regeneration.

#### 2.1.3. Decellularized Tissue-Based Materials

Decellularized tissue hydrogels represent a novel class of biomimetic materials. Their preparation involves removing cellular components from tissues or organs while preserving, as much as possible, the microstructure, biochemical composition, and bioactivity of the extracellular matrix (ECM). Such treatment minimizes immunogenicity while providing a native-like microenvironment that supports cell adhesion, proliferation, migration, and differentiation. The primary components—collagen, elastin, and glycosaminoglycans—help maintain the mechanical properties and viscoelasticity required for tissue function.

In recent years, hydrogels derived from decellularized tissues and their composites have demonstrated multiple therapeutic advantages in repairing radiation-induced skin injury. For example, acellular dermal matrix (ADM) hydrogels [103] significantly reduce wound area and radiation injury scores, decrease apoptosis, and increase epithelial thickness and hair follicle regeneration. Mechanistically, ADM hydrogels promote angiogenesis and vascular maturation, induce M2 macrophage polarization, and exert anti-inflammatory effects by downregulating IL-1β and IL-6 while upregulating IL-10. Composite systems incorporating human umbilical cord blood mesenchymal stem cells (hUCB-MSCs) with small intestinal submucosa (SIS)-derived ECM [104] further enhance the paracrine activity of MSCs, elevating HGF, VEGF, and ANG-1 secretion. These changes improve endothelial cell migration and angiogenesis, stimulate granulation tissue formation, and facilitate follicle regeneration in irradiated wounds. Likewise, Wharton’s jelly-derived MSCs (WJ-MSCs) [39] exhibit potent pro-repair effects by suppressing pro-inflammatory mediators (IFN, TNF, IL-1, IL-6) while upregulating pro-angiogenic and regenerative factors such as VEGF, EGF, bFGF, and KDR. This dual regulation activates angiogenic signaling pathways, accelerates endothelial proliferation and neovascularization, and promotes overall tissue regeneration. Different types of acellular tissue-based materials exhibit distinct structural features, signaling regulation, and repair mechanisms. Their representative characteristics are summarized in Table 4.

### 2.2. Synthetic Polymers and Their Composite Hydrogels

Synthetic polymer hydrogels are primarily constructed from artificially synthesized polymers or oligopeptides. Commonly used components include polyvinyl alcohol (PVA), polyacrylic acid (PAA), polyacrylamide (PAM), and their modified derivatives. These materials have been widely applied in tissue engineering and wound repair owing to their controllable composition, high batch consistency, tunable mechanical properties, and facile functional modification. As illustrated in Figure 6, synthetic polymers offer superior processability and reproducibility through molecular design of their structures, enabling adaptation to diverse tissue repair requirements [107].

In the context of RISI repair, synthetic hydrogels have demonstrated a spectrum of beneficial activities, including antioxidative, anti-inflammatory, and pro-angiogenic effects. For example, Yusen Hao et al. [108] developed an antioxidant heparin-mimetic peptide hydrogel (K16, KYKYEYEYAGEGDSS-4Sa), which effectively scavenges radiation-induced ROS, alleviates oxidative stress, and downregulates inflammatory responses, thereby creating a stable microenvironment for tissue repair. In addition, this hydrogel enhances angiogenesis and collagen deposition, accelerating tissue regeneration and remodeling.

Based on current research, various synthetic polymer materials exhibit distinct structural characteristics, reaction mechanisms, and repair outcomes. Through precise molecular design, these materials can integrate multiple therapeutic functions, including antioxidant activity, immune modulation, and tissue regeneration. Representative systems and their application characteristics are summarized in Table 5.

Materials based on polyacrylic acid and its derivatives have been extensively studied for RISI treatment. Kulshrestha et al. [117] synthesized a polyacrylic acid–sildenafil citrate hydrogel that significantly upregulated the fibroblast-specific marker α-SMA, indicating its role in fibroblast activation, granulation tissue formation, and improved wound repair. Congshu Huang et al. [116] designed a multifunctional ferulic acid (FA) hydrogel based on carbomer crosslinked polyacrylic acid. This hydrogel not only eliminated excess ROS, restored SOD activity, and maintained redox balance, but also inhibited NLRP3 inflammasome activation, reduced neutrophil infiltration, and alleviated inflammatory injury. Histological analyses revealed improved cutaneous blood flow, reduced edema and erythema, and more orderly collagen fiber deposition with restoration of epidermal structure. Mechanistically, FA hydrogel downregulated delayed-healing genes (IL-1A, IL-1B, MMP14), upregulated FGFR3, and activated the JAK/STAT and PI3K/AKT pathways, collectively regulating inflammation, angiogenesis, and tissue regeneration.

Beyond polyacrylic acid, other synthetic systems have also shown promise in RISI. Mingsheng Liu et al. [118] engineered a polyacryl-lysine (P-Ac-Lys) and polyvinyl alcohol–dihydroxyphenylalanine (PVA-DOPA) composite hydrogel (PAL@PVA-DOPA) loaded with lactic acid and pyruvic acid. Pyruvic acid, as a reducing molecule, scavenges ROS, mitigates DNA and protein damage, and improves cell survival, while lactic acid modulates the immune microenvironment by shifting macrophages from the pro-inflammatory M1 to the pro-repair M2 phenotype. In vivo experiments demonstrated that this hydrogel effectively reduced inflammatory infiltration, preserved tissue integrity, and accelerated healing of UV- and radiation-induced skin injuries.

### 2.3. Hybrid Hydrogels

With the integration of materials science and bioengineering, the design paradigm of hydrogels has evolved from single-polymer systems to multicomponent synergistic architectures. Unlike conventional structural hydrogels composed solely of natural or synthetic polymers, hybrid hydrogels integrate cells, exosomes, bioactive factors, nanoparticles, and functional polymers into hierarchically organized networks that couple biological activity with engineered functionality. These systems enable multilevel synergistic effects, including antioxidant protection, immune modulation, angiogenesis, and tissue remodeling, thereby exhibiting distinct advantages in the treatment of RISI.

#### 2.3.1. Cell and Exosome-Based Hybrid Systems

Cells and their secretory products have been widely utilized to construct hybrid hydrogels with intrinsic self-regulatory capabilities, owing to their natural regenerative and immunomodulatory potential. For instance, a human placental mesenchymal stem cell (hPMSC) gel [122] harnesses the paracrine functions of stem cells to suppress inflammation, upregulate the vascular marker CD31, and promote angiogenesis and tissue regeneration. Similarly, a GMSCs@pY hydrogel [123] encapsulating gingival mesenchymal stem cells (GMSCs) activates the EGFR–STAT3 signaling cascade, regulates downstream MMPs and BCL2, enhances keratinocyte proliferation, migration, and survival, and achieves dual anti-inflammatory and pro-angiogenic effects by downregulating IL-6, IL-1β, TNF-α and upregulating VEGF.

In exosome-integrated systems, the PDA-NPs@MSC-sEV hydrogel [124] exhibits multidimensional regulatory effects: it scavenges radiation-induced ROS/RNS and activates antioxidant enzymes (HO-1, CAT, SOD-1) to reduce DNA damage and apoptosis, while promoting the transition from the inflammatory to proliferative phase via downregulation of TNF-α, IL-6 and upregulation of IL-10, TGF-β. This system enhances angiogenesis, hair follicle regeneration, and collagen deposition.

#### 2.3.2. Bioactive Factors and Peptide-Driven Hybrid Systems

Regulating cellular behavior through biological signaling molecules represents another core strategy for hybrid hydrogel design. CSMP–PF hydrogel [125] co-loaded with TGF-β1 and the bioactive peptide SP establishes a multilevel regenerative microenvironment: TGF-β1 stimulates cell proliferation, migration, and ECM synthesis, while SP recruits bone marrow-derived MSCs and enhances myofibroblast activity. In animal models, this hydrogel markedly alleviated radiation- and infrared-induced loss of skin appendages and structural dermal damage, promoting synergistic lipid membrane and dermal regeneration.

In another example, a silybin–pomegranate oil nanocapsules hydrogel [126] combines natural anti-inflammatory agents with nanocarriers, effectively reducing inflammatory infiltration and early tissue injury, thereby exemplifying the integration of pharmacological and material-level regulation

#### 2.3.3. Nanoparticle and Inorganic Component-Based Hybrid Systems

Incorporating nanostructured elements into polymeric matrices can markedly enhance antioxidant capacity, mechanical strength, and functional stability without compromising biocompatibility.

The Nano-GDY@SH hydrogel [127], composed of sodium hyaluronate and graphdiyne (GDY) nanoparticles, forms a dual-protection system combining physical shielding and chemical antioxidation. This hybrid effectively reduces radiation energy deposition through its high water content while GDY scavenges free radicals to mitigate X-ray-induced DNA damage and cell inhibition.

The PHF@Res-2 hydrogel [128], integrating Pluronic F127-DA and HA-MA with resveratrol and Prussian blue nanoparticles (PPB), exhibits multidimensional synergy: it removes ROS, downregulates IL-6, IL-1β, TNF-α, and upregulates VEGF, CD31, and α-SMA, thereby promoting angiogenesis and tissue repair.

In a photoresponsive design, GelMA-based hybrid hydrogels [129] leverage curcumin–tannic acid nanoassemblies and near-infrared photothermal effects to scavenge radicals (DPPH, ABTS, •OH), suppress M1 inflammation, promote M2 polarization, and upregulate HSP90, CD31, and α-SMA, significantly improving skin perfusion and regeneration

#### 2.3.4. Biomimetic and Pathway-Regulated Hybrid Systems

Some hybrid hydrogels utilize biomimetic ECM strategies to achieve dual regulation of repair and signaling pathways. For instance, the oCP@As hydrogel [130], built on a glycopeptide framework with oxidized chondroitin sulfate, scavenges ROS, repairs DNA double-strand breaks, and activates RAD51-mediated homologous recombination, enhancing cell survival and radiation resistance. It further orchestrates cell proliferation, angiogenesis, and ECM remodeling through MAPK, Ras, Wnt, and TGF-β pathways, displaying multilevel signal-regulation capacity.

Similarly, GK@TAgel [131] targets mannose receptors via glycopeptide affinity to fine-tune the immune microenvironment, reduce pro-inflammatory cytokines, promote angiogenesis, and suppress abnormal epithelial proliferation, thereby accelerating RISI repair.

Overall, hybrid hydrogels achieve deep integration of material functionality and biological activity through multicomponent and multiscale design. Their strength lies in cross-domain synergy: nanoparticles and antioxidants provide physical and chemical protection, stem cells and exosomes modulate immune balance and angiogenesis, while signaling factors and bioactive molecules promote ECM remodeling and tissue regeneration. Such systemic designs enable hybrid hydrogels to deliver superior therapeutic efficacy in RISI repair compared with conventional natural or synthetic hydrogels. Representative hybrid systems and their molecular mechanisms are summarized in Table 6.

However, the incorporation of external functional components also introduces new challenges, including the size- and dose-dependent effects of nanoparticles, their potential biotoxicity and metabolic risks in vivo, and the limited stability and reproducibility of multicomponent systems during fabrication. Future research should emphasize precise interface engineering, dynamic component equilibrium, and long-term biosafety validation to achieve the integration of controlled release and visualized therapeutics. Through the deep convergence of materials science and bioengineering, hybrid hydrogels are poised to evolve into a next-generation platform for precision repair and personalized therapy in RISI.

### 2.4. Smart Hydrogels

In recent years, smart hydrogels have attracted growing attention in biomedical engineering owing to their sensitivity and controllability in response to external and internal stimuli. As shown in Figure 7, smart hydrogels can respond to diverse physical, chemical, and biological stimuli, enabling precise regulation and dynamic adaptation in applications such as tissue engineering and drug delivery [132].

In the treatment of RISI, such materials enable responsive regulation tailored to radiation-induced pathological changes. Building upon the advantages of conventional hydrogels, smart hydrogels introduce stimulus-responsiveness that allows structural or functional adjustments under specific triggering conditions. This capacity provides precise release and spatiotemporal control of therapeutic agents, thereby maintaining stable local drug concentrations, minimizing off-target side effects, and supporting multi-stage tissue repair [133].

One representative example is the DNAzyme hydrogel developed by Daijun Zhou et al. [134], which demonstrates intelligent responsiveness at the molecular level. In this design, a DNAzyme specifically targeting NLRP3 mRNA was encapsulated within ZIF-8 nanoparticles to improve stability, control release, and enhance tissue penetration. Further modification with a TAT transmembrane peptide increased cellular uptake. Under the acidic microenvironment of injured tissues, ZIF-8 dissociates, enabling on-demand DNAzyme release. Mechanistically, this system acts at multiple levels: at the molecular level, degrading NLRP3 mRNA and suppressing inflammasome activation; at the cellular level, reducing apoptosis and enhancing HaCaT cell migration and proliferation; and at the tissue level, attenuating inflammation, improving collagen deposition, and promoting follicle regeneration. Through the dual strategy of “environmental triggering + targeted nucleic acid regulation,” this hydrogel provides a novel paradigm for precision treatment of RISI.

In contrast to molecularly targeted approaches, Yi Xia et al. [135] designed an X-ray-responsive antioxidant hydrogel that exploits radiation as the stimulus. The hydrogel consists of a copolymer network of disulfide-containing hyperbranched poly (β-hydrazide ester) (PBAE), methacryloyl hyaluronic acid, and acrylamide. Upon exposure to radiation, oxidation of disulfide bonds disrupts the polymer network, triggering the release of encapsulated epidermal growth factor (EGF). This mechanism rapidly scavenges ROS, markedly reducing 8-OHdG levels (a marker of DNA oxidative damage), while enhancing fibroblast and keratinocyte proliferation and migration. In vivo studies further demonstrated that EGF loading significantly promoted angiogenesis as early as day 7, increasing vessel number and caliber, thereby accelerating tissue repair and improving healing quality. Table 7 summarizes the representative smart hydrogels discussed in this review, highlighting their structural characteristics, key biological functions, and current limitations.

Although research on smart hydrogels in RISI remains at an early stage, their unique advantages are becoming increasingly evident. With the continued convergence of polymer chemistry, nanomaterials, and bioengineering, these systems are expected to achieve more precise stimulus responsiveness and integrated multifunctionality—such as simultaneous antioxidant, immunomodulatory, and pro-angiogenic effects. Future efforts should prioritize addressing challenges in long-term safety, stability, and scalable production to accelerate the translation of smart hydrogels from laboratory studies to clinical application.

## 3. Conclusions

In summary, hydrogels, owing to their multifunctionality and structural designability, have become a focal point in the prevention and treatment of radiation-induced skin injury (RISI). From a material perspective, natural hydrogels—composed of biopolymers such as chitosan, hyaluronic acid, and gelatin—offer high biosafety, excellent biocompatibility, and the ability to mimic ECM. They exhibit potent therapeutic synergy when loaded with drugs or functionalized with bioactive molecules. In contrast, synthetic polymer hydrogels provide precise structural tunability and chemical stability. Through molecular design and functionalization of polymers such as PVA, PAA, and their derivatives, these systems effectively regulate oxidative stress, suppress inflammation, and promote tissue regeneration. The advent of hybrid and smart hydrogels has further expanded the therapeutic landscape. Hybrid hydrogels integrate nanoparticles, exosomes, and signaling molecules into multiscale composite architectures that enable synergistic coordination between material and biological signaling. Meanwhile, smart hydrogels offer stimuli-responsive drug release and spatiotemporal precision intervention in radiation-induced pathologies, laying the groundwork for personalized and dynamic therapeutic paradigms in future RISI management.

Despite these advances, most hydrogel-based strategies for RISI remain at the stage of cellular and animal studies. Although encouraging results have been reported in antioxidation, inflammation regulation, angiogenesis, and tissue regeneration, several challenges remain. First, the preparation and evaluation of hydrogels are largely confined to laboratory conditions, and their long-term stability, scalability, and controlled degradability require further validation in clinical settings [136]. Second, hybrid hydrogels may encounter barriers such as high costs, intricate fabrication processes, and incomplete regulatory frameworks during translation. For smart hydrogels, maintaining responsive sensitivity while ensuring biosafety and reproducibility remains a key challenge.

Future progress will depend not only on breakthroughs within individual disciplines but also on close interdisciplinary collaboration across materials science, radiation medicine, pharmacy, and bioengineering. Such integration will optimize material properties, deepen mechanistic understanding, and align innovations with clinical needs, ultimately accelerating the translation of laboratory findings into practice and providing more reliable and precise therapeutic solutions for RISI.

## Figures and Tables

**Figure 1 biomimetics-10-00758-f001:**
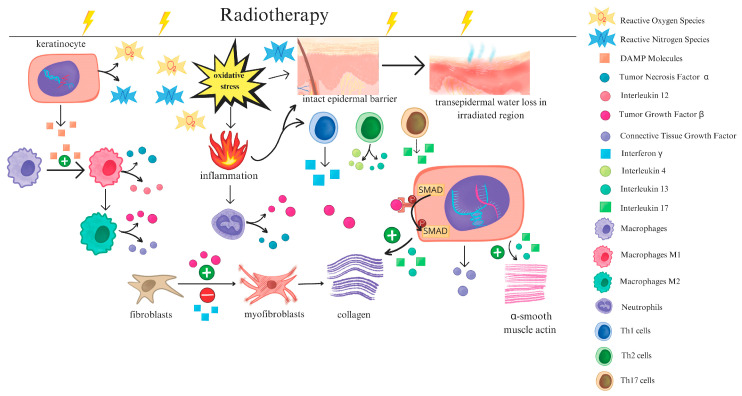
Schematic illustration of the subsequent alterations in immune cells, skin cells, and the skin barrier following radiotherapy. Reproduced from Bratborska et al. [13], under the CC BY 4.0 license.

**Figure 2 biomimetics-10-00758-f002:**
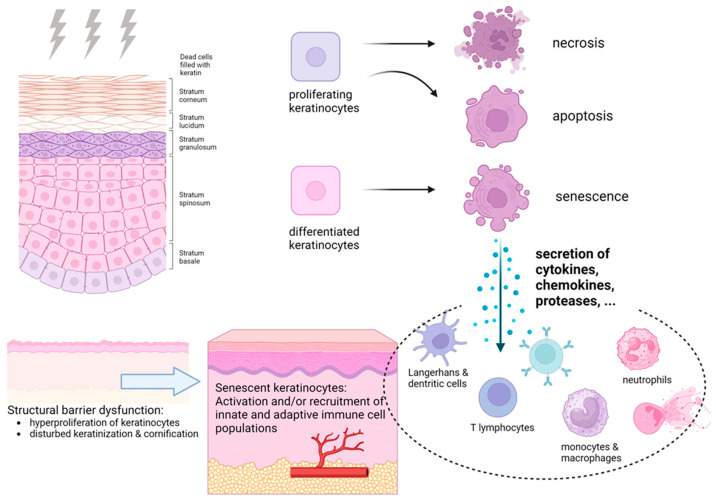
Pathophysiology of radiation-induced skin reactions. Reproduced from Rübe et al. [14], under the CC BY 4.0 license.

**Figure 3 biomimetics-10-00758-f003:**
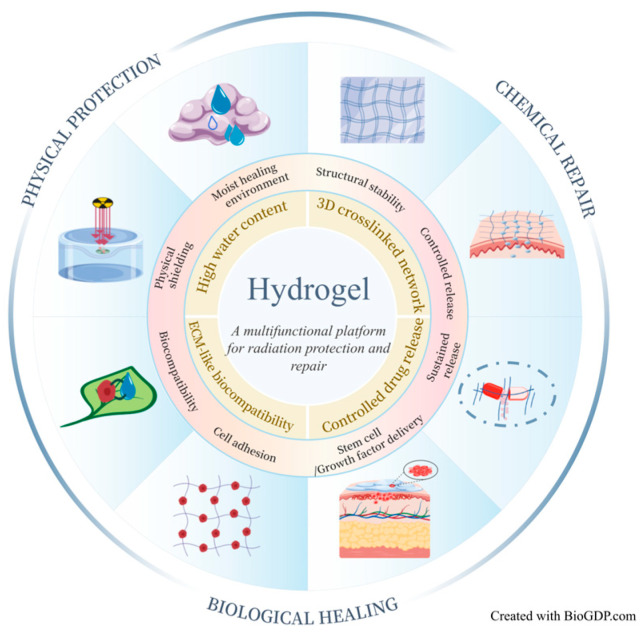
Schematic illustration of the multidimensional mechanisms of hydrogels in RISI protection and repair. Hydrogels act synergistically through multiple pathways—including physical shielding, chemical repair and biological healing—to establish an integrated therapeutic system that combines protection, drug delivery, and tissue regeneration. Created with BioGDP.com [49].

**Figure 4 biomimetics-10-00758-f004:**
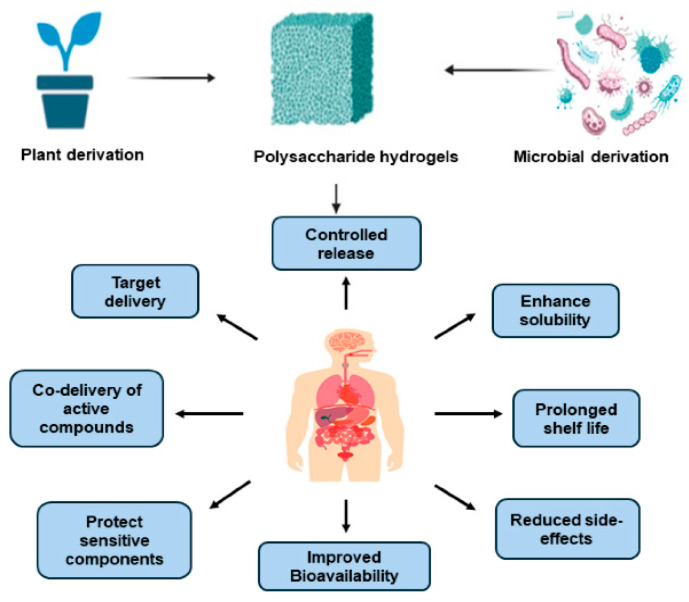
Sources and biological advantages of polysaccharide-based hydrogels. Reproduced from Sepe et al. [59], under the CC BY 4.0 license.

**Figure 5 biomimetics-10-00758-f005:**
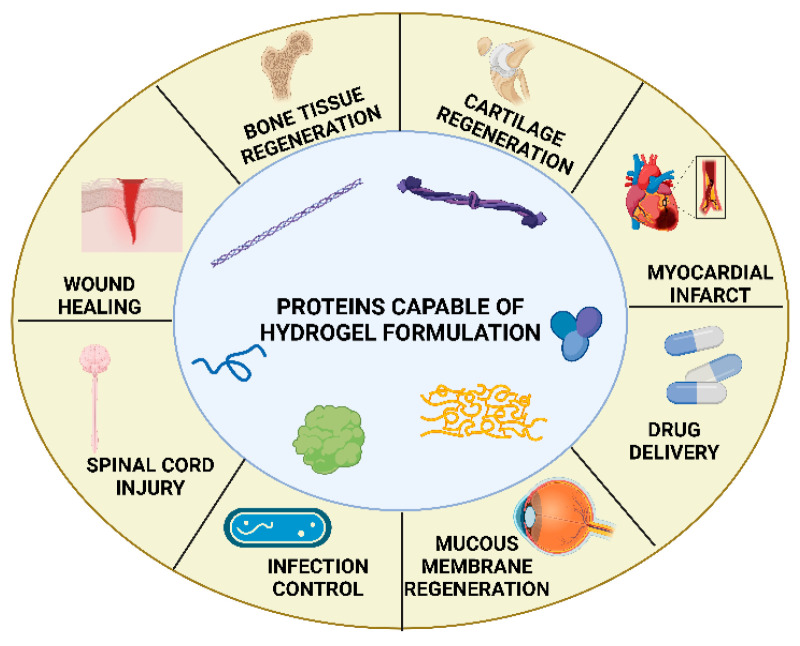
Typical proteins used for hydrogel construction and their multifield biomedical applications. Reproduced from Katona et al. [84], under the CC BY 4.0 license.

**Figure 6 biomimetics-10-00758-f006:**
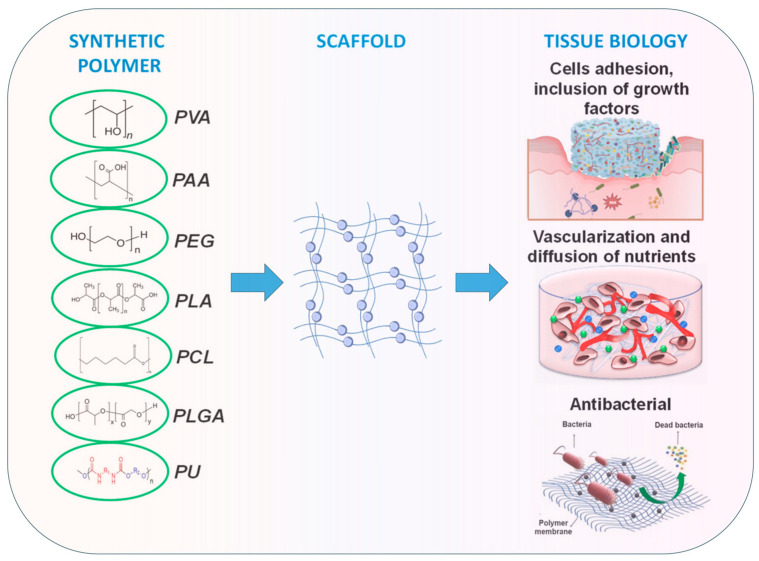
Synthetic polymers applied in wound healing and skin tissue engineering. Adapted from Avadanei-Luca et al. [107], under the CC BY 4.0 license.

**Figure 7 biomimetics-10-00758-f007:**
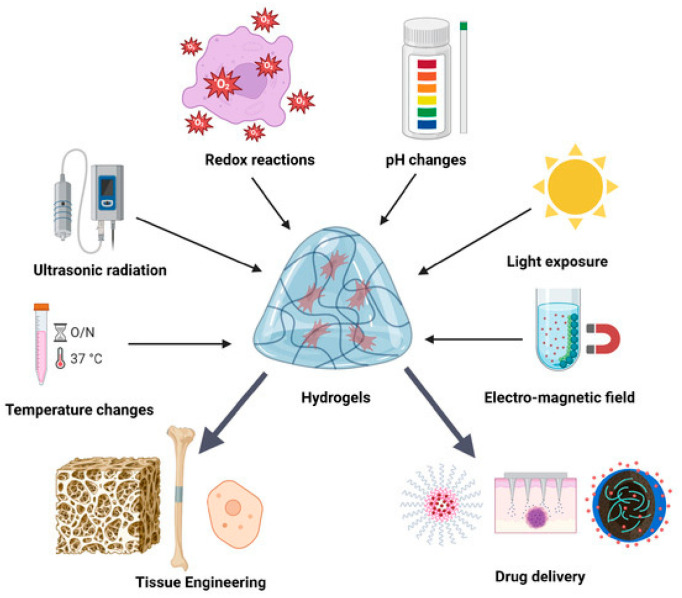
Schematic illustration of the regulation of smart hydrogel biological functions by various external stimuli. Reproduced from Munteanu et al. [132], under the CC BY 4.0 license.

**Table 1 biomimetics-10-00758-t001:** Current drugs and mechanism-oriented therapeutic strategies for RISI.

Type	Representative Drug/Strategy	Advantages	Limitations	References
Clinical Experience-Based Treatments	Pentoxifylline	Improves local blood flow and microcirculation Inhibits TGF-β-mediated fibrosis Combination with vitamin E alleviates post-radiation tissue sclerosis	Slow onset, requires long-term administration, limited efficacy in the acute phase	[21,22,23]
	RhEGF	Used in chronic RISI promotes epithelial regeneration and accelerates wound healing	Poor stability, short local retention, and relatively high cost	[24,25]
	ASCs	Applied in chronic RISI Abundant and easily obtainable source Secrete multiple growth factors (VEGF, FGF, EGF, TGF-β)	Complex culture process and high cost, limited clinical translation	[26,27,28]
	Topical Corticosteroids (e.g., mometasone furoate, betamethasone)	Used in acute RISI Suppress radiation-induced cytokine release (e.g., IL-6) Effectively reduce moist desquamation and delay grade III dermatitis	Long-term use damages the skin barrier, recurrence after withdrawal; mainly symptomatic, unable to prevent chronic fibrosis	[29,30,31,32]
	Emollients (e.g., triethanolamine cream)	Used in acute RISI Non-steroidal anti-inflammatory action, promotes macrophage recruitment and tissue repair Relieves dryness and pain	Primarily supportive, lacks targeted anti-inflammatory or anti-fibrotic effects; clinical efficacy remains controversial	[33,34,35]
Mechanism-Oriented Protective Strategies	Amifostine	Clinically approved radioprotective agent with well-established efficacy Scavenges ROS, suppress inflammation	Requires intravenous administration, dose-dependent systemic side effects	[36,37]
	MSCs	Exhibit radioresistance and maintain stemness after irradiation modulate the immune microenvironment via paracrine signaling suppress inflammation, promote angiogenesis and tissue regeneration improve RISI and enhance follicular and epithelial repair in animal models	Limited clinical data, mechanisms not fully elucidated; potential tumorigenic or tumor-protective risks; complex preparation, high cost, and challenges in GMP-compliant manufacturing and safety control	[38,39,40]

**Table 2 biomimetics-10-00758-t002:** Application characteristics of polysaccharide-based hydrogels and their derivatives in RISI repair.

Representative Material	Molecular/Structural Characteristics	Main Advantages	Key Mechanisms in RISI Repair	Limitations	References
Hyaluronic Acid 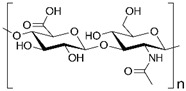	A linear glycosaminoglycan composed of D-glucuronic acid and N-acetyl-D-glucosamine.	Highly hydrophilic and biocompatible Easily modified and capable of loading bioactive molecules Structurally similar to the extracellular matrix (ECM), thereby promoting cell adhesion and migration	Scavenges ROS and alleviates oxidative stress, Upregulates VEGF expression to promote angiogenesis, Enhances fibroblast and hair follicle stem cell proliferation and differentiation.	Low mechanical strength Rapid enzymatic degradation Ph sensitivity Limited intrinsic bio-recognition sites	[60,61,62,63]
Alginate 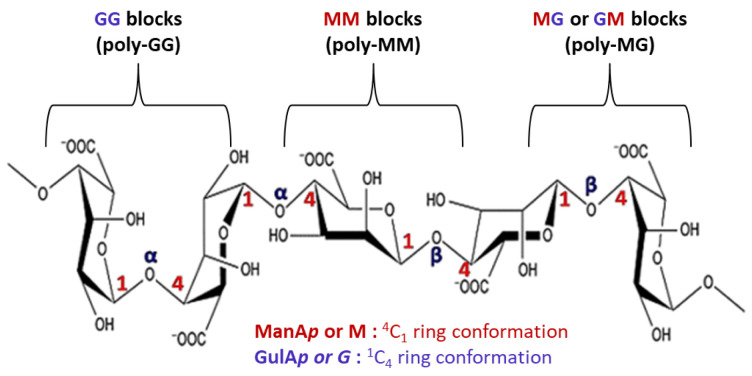 (image adapted from [64])	Composed of β-D-mannuronic acid (M) and α-L-guluronic acid (G) residues; cross-links with multivalent cations such as Ca^2+^ to form gel networks.	Readily available, low-cost, and rapidly gelling; exhibits high fluid absorption and excellent biocompatibility Easily combined with other polymers or functionalized	Ionic cross-linking forms a protective barrier, Scavenges ros and suppresses nlrp3 inflammasome activation, Enhances immune cell activity, Modulates TGF-β1 signaling and fibrosis progression.	Poor cell adhesion Brittleness Low cross-linking stability Prone to burst drug release	[65,66,67]
Chitosan 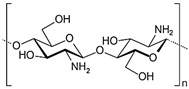	A cationic polysaccharide composed of β-(1→4)-linked D-glucosamine and N-acetyl-D-glucosamine units with variable deacetylation.	Exhibits antibacterial, hemostatic, and good film-forming properties readily complexes with anionic biomolecules biocompatible, biodegradable, and highly modifiable	Scavenges free radicals, promotes M2 macrophage polarization, inhibits MMP activity via MAPK signaling, facilitates angiogenesis and ECM remodeling.	Poor solubility (requires acidic environment) Limited mechanical strength Batch-to-batch variability.	[68,69,70,71,72]

**Table 3 biomimetics-10-00758-t003:** Application characteristics of protein-based hydrogels and their derivatives in RISI repair.

Representative Material	Molecular/Structural Characteristics	Main Advantages	Key Mechanisms in RISI Repair	Limitations	References
Gelatin 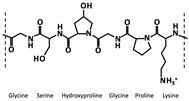 (image adapted from [86])	A collagen-derived polymer comprising repeating (Gly–XY) _n_ sequences, where X and Y are typically proline and hydroxyproline.	Excellent biocompatibility Contains cell-adhesion motifs Injectable and thermo-responsive Structurally similar to the ECM.	Enhances hemostasis, regulates inflammation, promotes angiogenesis and granulation, and drives collagen type III–I conversion.	Low mechanical strength Rapid enzymatic degradation Potential immunogenicity and batch variability.	[87,88,89,90]
Silk Fibroin 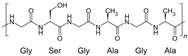	A fibrous protein derived from silkworm silk, rich in Gly/Ala/Ser residues and characterized by β-sheet crystalline domains.	High mechanical strength and thermal stability Tunable gelation and degradation rates Non-immunogenic and biocompatible	When combined with perfluorocarbon emulsions, alleviates hypoxia and promotes cell survival and tissue regeneration.	Complex preparation and high cost Low cell-encapsulation efficiency Batch-to-batch variability	[91,92,93,94]
Collagen 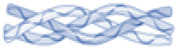	Predominantly type I and type III collagen composed of Gly–Pro–Hyp repeats forming a triple-helix structure.	High biocompatibility Excellent cell-adhesion capacity Multiple modifiable functional sites Provides strong structural support.	Enhances antioxidant capacity, reduces ROS accumulation, and promotes sustained anti-inflammatory effects and tissue repair.	Derived from animal tissues variability in purity and performance rapid degradation low drug-loading efficiency high production cost	[95,96,97,98]

**Table 4 biomimetics-10-00758-t004:** Application characteristics of acellular tissue-based hydrogels and their derivatives in RISI repair.

Material Category	Representative Material	Key Mechanisms in RISI Repair	Main Advantages	Limitations	References
Decellularized Tissue-Based Materials	ADM	Promotes angiogenesis and M2 macrophage polarization; downregulates IL-1β/IL-6 and upregulates IL-10	Preserves native ECM and signaling cues, Low immunogenicity, Combinable with MSCs to enhance regeneration.	Limited source availability High batch-to-batch variability Complex preparation Insufficient mechanical stability Restricted by regulatory constraints.	[39,103,104,105,106]
	SIS-ECM + hUCB-MSCs	Enhances HGF/VEGF/ANG-1 secretion; accelerates angiogenesis and granulation tissue formation			[107,108,109]
	WJ-MSCs + ECM	Downregulates IFN, TNF, IL-1, and IL-6; upregulates VEGF, EGF, bFGF, and KDR; activates angiogenic pathways; promotes endothelial proliferation and neovascularization.			[110,111,112]

**Table 5 biomimetics-10-00758-t005:** Application characteristics of synthetic polymer hydrogels and their derivatives in RISI repair.

Basic Material	Structural/Molecular Features	Main Advantages	Key Mechanisms in RISI Repair	Limitations	References
PVA 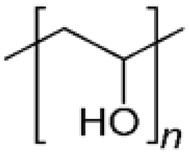	Linear polyvinyl polymer forming stable hydroxyl–crosslinked networks with polyphenols or polypeptides	Controllable composition Tunable mechanics High stability Easily functionalized	Scavenges ROS, alleviates DNA damage, modulates immune microenvironment, promotes M2 macrophage polarization.	Lacks biological cues Requires crosslinking or compositing	[109,110,111]
PAA 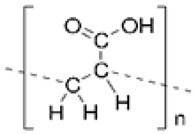	Linear polymer containing carboxyl groups; forms 3D crosslinked hydrogel networks	Excellent gelation High drug-loading capacity Precise controllability	Scavenges ROS, restores SOD activity, inhibits NLRP3 inflammasome activation, modulates JAK/STAT and PI3K/AKT pathways, enhances fibroblast activation and granulation.	Difficult to control degradation Low mechanical strength Needs reinforcement	[112,113,114,115,116,117]
PAM 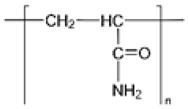	Acrylamide-based polymer crosslinked or copolymerized with functional monomers	Highly designable Compatible with drug or signal molecule loading	Removes ROS, reduces inflammation, promotes ECM remodeling and angiogenesis.	Biologically inert requires chemical modification	[118,119,120,121]
Functional synthetic peptides (e.g., K16)	Self-assembling heparin-mimicking peptide (KYKYEYEYAGEGDSS-4Sa) forming ECM-like networks	High functionalization potential, Mimics ECM, Adsorbs inflammatory cytokines	Scavenges radiation-induced ROS, suppresses inflammation, promotes angiogenesis and collagen deposition.	High cost Low stability Degradation profile requires optimization.	[108]
Composite synthetic polymers (e.g., P-Ac-Lys/PVA-DOPA)	Copolymer hydrogel combining poly(acryloyl-lysine) and PVA-DOPA; capable of loading lactate/pyruvate	Multifunctional synergy: antioxidant, immunomodulatory, and tissue-protective	Pyruvate scavenges ROS; lactate promotes M1→M2 macrophage polarization, improves immune balance, and accelerates tissue repair	Complex synthesis limited clinical validation	[118]

**Table 6 biomimetics-10-00758-t006:** Molecular basis and functional characteristics of hybrid hydrogels in RISI repair.

Type of Hybrid System	Representative Materials/Structural Features	Main Advantages	Key Mechanisms in RISI Repair	Limitations	References
Cell-Based Hybrids	hPMSC Gel GMSCs@pY hydrogels encapsulating MSCs for paracrine regulation	Integrate cellular bioactivity with material functionality, Enable cross-scale signaling and biomimetic microenvironment reconstruction.	Upregulate VEGF, CD31; downregulate IL-6, IL-1β, TNF-α, activate EGFR–STAT3.	Limited cell viability and reproducibility, high batch variability, complex system design.	[123,126]
Exosome/Nanoparticle Hybrids	PDA-NPs@MSC-sEV dopamine nanoparticles combined with MSC-derived small extracellular vesicles	Achieve multi-mechanistic synergy of antioxidant, immunomodulatory, and regenerative effects with high structural stability.	Scavenges ROS/RNS, upregulates HO-1, CAT, SOD-1,modulates cytokines (↓TNF-α, IL-6; ↑IL-10, TGF-β), promotes angiogenesis, follicle regeneration, and collagen remodeling.	Complex exosome isolation, limited long-term stability and scalability.	[124]
Bioactive-Factor/Peptide-Driven Systems	CSMP-PF (dual-signal peptide hydrogel co-loading TGF-β1/SP) Silybin–Pomegranate Oil Nanocapsule Hydrogel (bioactive molecule–nanocarrier network)	Enable multi-signal regulation and layered tissue repair via sustained release of active factors.	TGF-β1 promotes ECM synthesis, SP recruits MSCs and activates myofibroblasts, reduces inflammatory infiltration, restores lipid membrane and dermal architecture.	Poor growth-factor stability, uncontrolled release kinetics; potential immunogenicity.	[122,125]
Nano/Inorganic Hybrids	Nano-GDY@SH (HA-graphdiyne) PHF@Res-2 (dual-network hydrogel with antioxidant nanoparticles) GelMA-Cur/TA (photothermal-responsive nanocomposite)	Provide dual physical–chemical protection, integrates photothermal immunoregulation with tissue repair, exhibits high mechanical strength.	Scavenges ROS, •OH; downregulates IL-6, IL-1β, TNF-α; upregulates VEGF, α-SMA; promotes M2 polarization and angiogenesis; photothermal activation of HSP90, CD31 enhances perfusion	Photothermal response requires external stimuli; limited tissue penetration; complex synthesis; uncertain long-term nanomaterial safety	[127,128,129]
Biomimetic/Signal-Regulatory Systems	oCP@As (glycopeptide-oxidized chondroitin sulfate biomimetic system) GK@TAgel (glycopeptide-receptor-targeted hydrogel)	Mimics ECM architecture and enables multidimensional pathway regulation, restores immune balance and supports regeneration.	Scavenges ROS, repairs DNA breaks via RAD51-mediated homologous recombination; modulates MAPK, Ras, Wnt, TGF-β signaling; promotes angiogenesis and ECM remodeling	Requires high fabrication precision; limited translational validation	[130,131]

**Table 7 biomimetics-10-00758-t007:** Structural and functional characteristics of smart hydrogels in radiation-induced skin injury (RISI) repair.

Type of Responsive System	Representative Material	Structural Characteristics	Main Advantages	Key Mechanisms in RISI Repair	Limitations	References
Acidic Microenvironment-Responsive System	DNAzyme@ZIF-8/TAT hydrogel	DNAzyme encapsulated within ZIF-8 nanocarriers, TAT peptide facilitates cellular penetration and targeted delivery.	Enables molecularly targeted, PH-triggered on-demand release Achieves spatiotemporal control with integrated anti-inflammatory, anti-apoptotic, and reparative effects	Acidic ph degrades ZIF-8 to release DNAzyme, Suppresses NLRP3, reduces apoptosis, Promotes HaCaT migration and collagen regeneration.	Complex synthesis poor nucleic acid stability	[134]
Radiation-Triggered Responsive System	PBAE/HA-MA/AM-EGF hydrogel	Hydrogel network containing disulfide bonds and EGF, irradiation induces bond cleavage and factor release.	Radiation-adaptive response enables rapid release combines antioxidative and pro-angiogenic functions	Radiation cleaves disulfide bonds to release EGF, scavenges ROS, lowers 8-OHdG, promotes fibroblast and keratinocyte proliferation.	Dose-dependent response limited biosafety data	[135]

## Data Availability

No new data were created or analyzed in this study. Data sharing is not applicable to this article.

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
