# Peer review of "Hydrogel-Based Strategies for the Prevention and Treatment of Radiation-Induced Skin Injury: Progress and Mechanistic Insights"

_biomimetics, 2025, doi:10.3390/biomimetics10110758_

Round 1

Reviewer 1 Report

Comments and Suggestions for Authors

General comments

One of the most common complications of radiation therapy is radiation-induced skin injury (RISI), which significantly impairs patients' quality of life and has long-term adverse physical and psychosocial effects. RISI can manifest as erythema, edema, pigment changes, and varying degrees of scaling and is often characterized by a stepwise progression, with the process evolving from acute inflammation to chronic oxidative damage and fibrosis, which compromises the structural stability and regenerative capacity of the skin. Modern wound dressings that utilize the structural and functional versatility and multifunctionality of hydrogels can significantly aid in the treatment of RISI. This peer-reviewed manuscript describes the potential applications of hydrogel wound dressings, the range of which has significantly expanded in recent years thanks to molecular engineering and hybridization of hydrogel materials.

It should be noted that the topic of the study is highly relevant, so the usefulness of this peer-reviewed paper is beyond doubt. While the literature contains good review articles on the medical aspects of hydrogel use, this review does not duplicate those and may be useful to a wider audience. However, the presentation of the material is not very appealing. The manuscript does not contain a single figure or diagram that would engage readers. The work contains only one table, which is also inappropriate, as tables help classify the material. Furthermore, the small number of references (69) does not indicate a thorough review of the material. Similar articles containing numerous figures and tables could be selected as examples of proper presentation:

  1. Avadanei-Luca, S.; Nacu, I.; Avadanei, A.N.; Pertea, M.; Tamba, B.; Verestiuc, L.; Scripcariu, V. Tissue Regeneration of Radiation-Induced Skin Damages Using Protein/Polysaccharide-Based Bioengineered Scaffolds and Adipose-Derived Stem Cells: A Review. Int. J. Mol. Sci. 2025, 26, 6469. https://doi.org/10.3390/ijms26136469 326 references, 11 figures, 5 tables
  2. Katona, G.; Sipos, B.; Csóka, I. Advancements in the Field of Protein-Based Hydrogels: Main Types, Characteristics, and Their Applications. Gels 2025, 11, 306. https://doi.org/10.3390/gels11050306 131 references, 7 figures, 7 tables

The work is not yet ready for publication and needs to be significantly revised:

  1. Several color illustrations should be added to the text of the manuscript. The article is read with interest if it is well illustrated. In particular, figures may be taken from other MDPI articles with a mandatory reference to the article containing the figure used.
  2. Tables similar to those given in [1,2] should be added to the text, since tables help to classify the material and link it to specific literary sources.
  3. Few references to literature sources, their number (69) does not correspond to a good review. There are a lot of articles on various hydrogel wound dressings. A good review contains more than 120 references.

Specific comments

  1. The titles of sections 2.1.2 and 2.1.3 Protein-Based Materials are the same, although section 2.1.3 discusses decellularized tissue-derived hydrogels.
  2. The letter "C" is missing from the title of section 2.1 Natural-Origin and their composite Hydrogels.

Author Response

Comments 1: Several color illustrations should be added to the text of the manuscript. The article is read with interest if it is well illustrated. In particular, figures may be taken from other MDPI articles with a mandatory reference to the article containing the figure used.

Response 1: Thank you for your valuable suggestions. We fully agree that the inclusion of color illustrations enhances both the clarity and overall appeal of the manuscript. Accordingly, several color figures have been added to better visualize the structures and mechanisms of hydrogels in RISI repair. Specifically, Figures 1–7 were introduced to illustrate key mechanisms and material classifications, and a graphical abstract has been added as requested by the editors. In addition, material structure diagrams have been incorporated into Tables 2, 3, and 5 to improve visual understanding.

All revised illustrations are accompanied by appropriate permissions and citations (see Pages 2–4, 6, 8, 9, 11, 12, 15, and 22).

The updated figure titles and sources are clearly marked in red in the revised manuscript.

Comments 2: Tables similar to those given in [1,2] should be added to the text, since tables help to classify the material and link it to specific literary sources.

Response 2: Thank you for this valuable recommendation. Following your suggestion, we have added seven new tables to improve classification and clarity (see Tables 1–7, pp. 4–5, 8-9, 11-13, 15-16, 18-19, 23).

We carefully examined the two recommended review articles before citation:

(i) Katona G. et al., Gels 2025, 11, 306.

This article systematically summarizes the main types, structural features, and biomedical applications of protein-based hydrogels, highlighting their mechanical challenges and hybridization strategies. It directly informed our discussion on protein-derived materials (Figure 5) and helped refine our comparison between natural and synthetic hydrogel frameworks.

(ii) Avadanei-Luca S. et al., Int. J. Mol. Sci. 2025, 26, 6469.

This paper provides a comprehensive discussion of protein/polysaccharide-based scaffolds combined with adipose-derived stem cells for the repair of radiation-induced skin injury. It offers valuable mechanistic insight into how stem-cell-loaded bioengineered hydrogels regulate angiogenesis, inflammation, and collagen remodeling, which supports and contextualizes our section on synthetic and composite hydrogel systems (Figure 6). We cited it to emphasize the translational potential of engineered scaffolds in RISI-related tissue regeneration.

Both papers complement our review by broadening its scope and enhancing the linkage between material design and biological performance. Their integration strengthens the scientific foundation of our manuscript without overlapping with our own analytical focus.

All corresponding figure titles and adapted sources have been clearly marked in red in the revised version. Deleted parts are indicated in blue and marked with a strikethrough.

Comments 3: Few references to literature sources, their number (69) does not correspond to a good review. There are a lot of articles on various hydrogel wound dressings. A good review contains more than 120 references.

Response 3: We appreciate the reviewer’s insightful comment. In response, we have substantially expanded the reference list from 67 to 139 citations, ensuring broader and more comprehensive coverage of relevant literature.

Comments 4: The titles of sections 2.1.2 and 2.1.3 Protein-Based Materials are the same, although section 2.1.3 discusses decellularized tissue-derived hydrogels.

Response 4: Thank you for catching this oversight. We have corrected the section title 2.1.3 to accurately reflect its content. The revised title now reads:

“2.1.3 Decellularized Tissue–Based Materials”

This modification ensures consistency between the title and the section content (see Page 12, Line 345).”

Comments 5: The letter "C" is missing from the title of section 2.1 Natural-Origin and their composite Hydrogels

Response 5: We appreciate the reviewer for noting this typographical error. The section title has been corrected to:

“2.1 Natural-Origin and Their Composite Hydrogels”

The correction has been implemented in the revised manuscript (see Page 7, Line 200).

4. Response to Comments on the Quality of English Language

Point 1:

Response 1: Thank you for the positive assessment of the English language. Although no major issues were identified, minor stylistic and grammatical refinements were made to further enhance clarity and precision in the revised manuscript.

5. Additional clarifications

We would like to sincerely thank the reviewers and editors once again for their valuable feedback. All comments and suggestions have been carefully addressed, leading to substantial improvements in both the content and presentation of the manuscript. We believe these revisions have significantly enhanced the overall quality and scientific rigor of the work.

With the editor's permission, we have also revised the author list and project funding information. The author change form has been submitted as a separate attachment.

Reviewer 2 Report

Comments and Suggestions for Authors

This manuscript summarized the recent progress of radiation-induced skin injury (RISI) treatment by hydrogel strategies. Overall, the manuscript is quite comprehensive, focusing on the different types of hydrogels in recent research, including nature-sourced hydrogels, synthetic hydrogels, and composite hydrogels. The most notable shortcoming is the absence of schematic diagrams, and the structure is not particularly clear. The following are the suggestions for the author’s reference.

  1. What are the keys for the RISI treatment in clinic? And why hydrogel strategies are important, unique and useful compared to the clinical methods? Hydrogels have been widely used in clinical practice as wound dressings, so does RISI has any hydrogel products? Are there any requirements for the material's degradability, mechanical properties, etc.?
  2. A schematic diagram should be added to simply describe the progress and mechanism after the injuries are treated with hydrogel materials. In addition, for the cited literatures, it’s recommended to incorporate some diagrams from the referenced studies.
  3. Basically, the hydrogels can provide a moist environment and physical barriers, as well incorporated with some additional therapeutic agents, which are easily to understand. For the composite hydrogel or hybrid hydrogel, most studies emphasized the effects of certain “components” such as bioactive factors, nanoparticles, etc. Therefore, the “other composite hydrogels” and “hybrid hydrogels” are essentially no different. The author may consider combine these 2 sections together.
  4. In order to provide clearer mentoring for researchers, authors can summarize the specific functions of each material. For example, the different polysaccharides itself can form highly hydration environment with grate biocompatibility, and different charges materials can be chosen for different purpose. Therefore, for Table 1, it is suggested to summarize the materials’ molecular structure, characteristics (including advantages and disadvantages), and key functions for RISI. The current table is quite disorganized, almost a list of cited references while losing citation numbers, and it’s hard to read.
  5. Protein-based materials can have special hemostatic function, the mechanism (structure-function) should be illustrated.
  6. For external composition, what are the disadvantages or challenges for them? For example, the incorporated nanoparticles, are there requirements for the size, concentration, toxicity, or release properties?
  7. The author should carefully check the writing, for example “2.1. Natural-Origin and their omposite Hydrogels”, 2.1.2 and 2.1.3 have the same title.

Author Response

Comments 1: What are the keys for the RISI treatment in clinic? And why hydrogel strategies are important, unique and useful compared to the clinical methods? Hydrogels have been widely used in clinical practice as wound dressings, so does RISI has any hydrogel products? Are there any requirements for the material's degradability, mechanical properties, etc.?

Response 1:

I will explain this in four main points:

(i) Major Clinical Challenges in RISI Treatment

Clinical management of radiation-induced skin injury (RISI) requires simultaneous regulation of inflammation, oxidative stress, and tissue remodeling. Conventional interventions—including topical corticosteroids, emollients, pentoxifylline, and growth factors—provide only temporary symptom relief and suffer from short drug retention, systemic side effects, and limited control over the local microenvironment. These limitations underscore the need for multifunctional, localized strategies that can protect, repair, and regenerate irradiated skin.

(ii) Distinct Advantages of Hydrogel-Based Approaches

With an increasing understanding of RISI’s molecular mechanisms, therapeutic research is shifting from nonspecific symptomatic management toward mechanism-driven radioprotective development. Hydrogels, characterized by high water content, soft viscoelasticity, and tunable physicochemical properties, provide a versatile therapeutic platform. They can prolong skin adherence, enable sustained drug release, modulate the wound microenvironment (through pH, ROS, or enzymatic responsiveness), and mimic the extracellular matrix (ECM).

Accordingly, hydrogels serve not only as radioprotective barriers but also as therapeutic carriers, integrating protective and regenerative functions within a single system. These combined attributes position hydrogels as promising candidates for next-generation radioprotective biomaterials.

(iii) Clinical Translation and Current Applications

Although no hydrogel formulation has yet been approved specifically for RISI, several clinically available wound dressings—including sodium hyaluronate gels, chitosan–gelatin hydrogels, and EGF- or DFO-loaded composites—have been repurposed for radiation-injury management with encouraging preclinical and early clinical results.

Ongoing investigations are also exploring radioprotective hydrogel patches for prophylactic use during radiotherapy, aiming to mitigate acute skin reactions and accelerate post-irradiation healing.

(iv) Material Design Considerations

An ideal hydrogel for RISI therapy should exhibit the following essential characteristics:

Degradability: It should degrade progressively in synchrony with the staged wound-healing process (hemostasis inflammation proliferation remodeling), avoiding premature collapse or long-term persistence at the wound site. Degradation byproducts must be non-toxic and pH-neutral. Rapidly degradable systems are suitable for acute lesions requiring short-term drug delivery, whereas slower degradation benefits chronic RISI, ensuring sustained antioxidant and anti-inflammatory effects.

Mechanical properties: The hydrogel should be soft, flexible, and conformable, adhering closely to skin contours and movement without impeding reepithelialization. It must also possess adequate toughness to withstand dressing changes and patient motion, while maintaining structural integrity in moist or exudative conditions.

Porosity, mesh size, and swelling behavior: The network architecture should enable oxygen and nutrient diffusion while regulating drug-release kinetics, determined by crosslinking density and average molecular weight between crosslinks. These parameters can be tuned to achieve the desired release profile. A high water content helps maintain a moist healing microenvironment and promotes exudate absorption, though excessive swelling should be avoided to prevent skin maceration. Swelling behavior can be precisely adjusted by controlling the hydrophilic/hydrophobic ratio and crosslinking density.

Biocompatibility and safety: The material must exhibit low immunogenicity, minimal endotoxin contamination, and allow for sterilization without compromising physicochemical integrity. A clean degradation profile is critical to avoid acidic or reactive byproducts that could aggravate inflammation in irradiated tissue. Potential allergenicity associated with certain natural polymers should also be carefully evaluated during material selection.

Comments 2: A schematic diagram should be added to simply describe the progress and mechanism after the injuries are treated with hydrogel materials. In addition, for the cited literatures, it’s recommended to incorporate some diagrams from the referenced studies.

Response 2:  Thank you for your constructive suggestions. To enhance visual clarity, we have added two new diagrams—a Graphical Abstract (page 1) and Figure 3 (page 6)—that summarize the types and underlying mechanisms of hydrogel-based treatments for radiation-induced skin injury (RISI).

In addition, several illustrations adapted from referenced studies have been incorporated, with appropriate source attributions and citations provided (see Figures 1–2, 4–7; pages 3–4, 8, 11, 15, and 22). All newly added or modified figures are highlighted in red in the revised manuscript. .

Comments 3: Basically, the hydrogels can provide a moist environment and physical barriers, as well incorporated with some additional therapeutic agents, which are easily to understand. For the composite hydrogel or hybrid hydrogel, most studies emphasized the effects of certain “components” such as bioactive factors, nanoparticles, etc. Therefore, the “other composite hydrogels” and “hybrid hydrogels” are essentially no different. The author may consider combine these 2 sections together.

Response 3: We appreciate the reviewer’s insightful observation. After careful consideration, we have merged the sections on “Other Composite Hydrogels” and “Hybrid Hydrogels” into a unified section entitled “Hybrid Hydrogels” (see Section 2.3, pp. 1619).

This integration eliminates redundancy and better reflects the multicomponent design concept, where nanoparticles, bioactive factors, exosomes, and signaling molecules synergistically contribute to multiscale protection and tissue regeneration. The corresponding tables and references have been updated accordingly. (pp. 1819).

Comments 4: In order to provide clearer mentoring for researchers, authors can summarize the specific functions of each material. For example, the different polysaccharides itself can form highly hydration environment with grate biocompatibility, and different charges materials can be chosen for different purpose. Therefore, for Table 1, it is suggested to summarize the materials’ molecular structure, characteristics (including advantages and disadvantages), and key functions for RISI. The current table is quite disorganized, almost a list of cited references while losing citation numbers, and it’s hard to read.

Response 4: Thank you for your valuable suggestions. Table 1 has been completely reorganized and divided into Tables 2–4 to improve readability and provide clearer guidance for researchers. The revised tables now summarize, for each material, its molecular or structural properties, key advantages and disadvantages, and principal mechanisms/functions in RISI repair, with complete and updated reference numbers.

This structural reorganization facilitates direct comparison among different polysaccharide-based hydrogels and clarifies the relationship between specific molecular features (e.g., charge, hydration capacity, and functional groups) and their corresponding biological functions. The updated tables (pages 8-9, 11-13) are highlighted in red in the revised manuscript.

Comments 5: Protein-based materials can have special hemostatic function, the mechanism (structure-function) should be illustrated.

Response 5: Thank you for this valuable suggestion, reviewer. The hemostatic properties of protein-based materials are indeed an advantage of protein-based hydrogels. We have added a new table (Table 3, pages 11-12) highlighting these hemostatic advantages. We have also provided a supplementary description of their hemostatic mechanisms (structure-function) in the text.

Comments 6: For external composition, what are the disadvantages or challenges for them? For example, the incorporated nanoparticles, are there requirements for the size, concentration, toxicity, or release properties?

Response 6: Thank you for raising this important point. We have added a new table (Table 6) in Section 2.3 (pp. 18-19) discussing the application characteristics of hydrogels containing external functional components.

Nanoparticle size, dosage, and surface chemistry have a crucial impact on biodistribution, biocompatibility, and release kinetics. Concentration-dependent cytotoxicity and potential long-term metabolic risks must be carefully assessed. Furthermore, the reproducibility and stability of multicomponent hybrid systems remain major challenges in translational research.

In addition, the Conclusion section has been expanded to emphasize the need for precise interface control, dynamic component balance, and long-term safety validation to ensure clinical applicability. These revisions are highlighted in red in the revised manuscript.

Comments 7: The author should carefully check the writing, for example “2.1. Natural-Origin and their omposite Hydrogels”, 2.1.2 and 2.1.3 have the same title.

Response 7: We sincerely thank the reviewer for this careful observation. The typographical error in “2.1. Natural-Origin and their omposite Hydrogels” has been corrected to “2.1. Natural-Origin and Composite Hydrogels.” In addition, the duplicate section titles for 2.1.2 and 2.1.3 have been revised to clearly distinguish their contents:

2.1.2 Protein-Based Materials, and

2.1.3 Decellularized Tissue–Based Materials.

All corresponding section titles and references have been updated accordingly in the revised manuscript.

4. Response to Comments on the Quality of English Language

Thank you for this observation. The entire manuscript has been carefully revised for English grammar, clarity, and style. We have polished sentence structures, refined terminology, and improved overall readability to ensure the manuscript meets high academic standards.

All revised sections have been marked in red in the re-submitted file. Deleted parts are indicated in blue and marked with a strikethrough.

5. Additional clarifications

We would like to sincerely thank the reviewers and editors once again for their valuable feedback. All comments and suggestions have been carefully addressed, leading to substantial improvements in both the content and presentation of the manuscript. We believe these revisions have significantly enhanced the overall quality and scientific rigor of the work.

With the editor's permission, we have also revised the author list and project funding information. The author change form has been submitted as a separate attachment.

Round 2

Reviewer 1 Report

Comments and Suggestions for Authors

The authors have done a great job of addressing the aforementioned shortcomings and improving the manuscript. I believe the revised version is suitable for publication.
